# Towards Adaptive ML Benchmarks: Web-Agent-Driven Construction, Domain Expansion, and Metric Optimization

## Abstract

Recent advances in large language models (LLMs) have enabled the emergence of general-purpose agents for automating end-to-end machine learning (ML) workflows, including data analysis, feature engineering, model training, and competition solving. However, existing benchmarks remain limited in task coverage, domain diversity, difficulty modeling, and evaluation rigor, failing to capture the full capabilities of such agents in realistic settings. We present TAM Bench, a diverse, realistic, and structured benchmark for evaluating LLM-based agents on end-to-end ML tasks. TAM Bench features three key innovations: (1) A browser automation and LLM-based task acquisition system that automatically collects and structures ML challenges from platforms such as Kaggle, AIcrowd, and Biendata, spanning multiple task types and data modalities (e.g., tabular, text, image, graph, audio); (2) A leaderboard-driven difficulty modeling mechanism that estimates task complexity using participant counts and score dispersion, enabling scalable and objective task calibration; (3) A multi-dimensional evaluation framework incorporating performance, format compliance, constraint adherence, and task generalization. Based on 150 curated AutoML tasks, we construct three benchmark subsets of different sizes—Lite, Medium, and Full—designed for varying evaluation scenarios. The Lite version, with 18 tasks and balanced coverage across modalities and difficulty levels, serves as a practical testbed for daily benchmarking and comparative studies.

## 1 Introduction

With the successful application of large language models (LLMs) in areas like code generation and task planning, an increasing number of researchers have begun exploring their potential in automating end-to-end machine learning (ML) tasks. These tasks include model training, feature engineering, data analysis, and competition problem solving. This emerging direction has led to the development of intelligent agent systems for the data science pipeline, such as AIDEJiang et al. (2025) AutoMindOu et al. (2025) ML-AgentLiu et al. (2025b) and ML-MasterLiu et al. (2025a). These systems aim to build general-purpose agents capable of task understanding, tool invocation, and workflow execution to fully automate the ML pipeline.

However, the current evaluation of LLMs' comprehensive capabilities in ML tasks remains insufficient. Existing benchmarks suffer from the following major issues: 1) High Manual Cost of Data Collection,Traditional benchmarks rely heavily on manual efforts for competition collection, content extraction, and task filtering. While this ensures data quality, it is inefficient, costly, and difficult to scale.2) Imbalanced Distribution of Task Types and Application Domains, Existing benchmarks are biased in both task types and application scenarios. For example, MLEBenchChan et al. (2025) includes 36 image/text classification tasks but only sparsely covers tasks like object detection or image-text generation. CALMFeng et al. (2024) focuses exclusively on tabular modeling in the finance domain. Common real-world applications such as e-commerce recommendation and educational analytics are missing, which limits the representativeness and generalizability of evaluations.3) Unreasonable Task Difficulty Modeling, In a diverse task set, reasonable difficulty grading is essential to assess agents fairly. It helps highlight performance gaps across complexity levels and

understand generalization boundaries. Currently, only MLEBenchChan et al. (2025) attempts difficulty modeling via expert-annotated time estimates, which is subjective, costly, and not scalable. 4) Single-Dimensional Evaluation Metrics, Most benchmarks rely on single metrics, e.g., medals of MLEBenchChan et al. (2025) or performance improvement over baseline of MLAgentBenchHuang et al. (2024). This oversimplifies agent assessment and may cause reward hacking, where agents prioritize scores while ignoring format or business constraints. To address these limitations, we propose TAM-Bench—a structured, diverse, and realistic benchmark to evaluate the comprehensive capabilities of LLMs in ML tasks. Key innovations include:

- **Automated Task Collection and Standardization across Domains:** Building on MLEBenchChan et al. (2025), we identify key ML task types and domains. Inspired by MCPHou et al. (2025) and Browser-Use, we develop a web-agent-based system to scrape, extract, and schema-unify tasks from Kaggle, AIcrowd, Biendata, etc., enabling scalable cross-domain task acquisition and reducing manual overhead.

- **Leaderboard-Based Difficulty Modeling:** Using real competition data (e.g., participant count and score distribution), we design an automated and extensible task difficulty scoring system, reducing subjectivity and improving scalability.

- **Multi-Dimensional Evaluation Framework:** We incorporate performance, format compliance, constraint adherence, and generalization to provide a holistic agent capability assessment. Our design prevents reward hacking by ensuring improvements follow real-world constraints.

TAM-Bench aims to enable robust evaluation and development of LLM-powered agents in complex, realistic ML workflows, establishing a key standard for assessing data science capabilities.

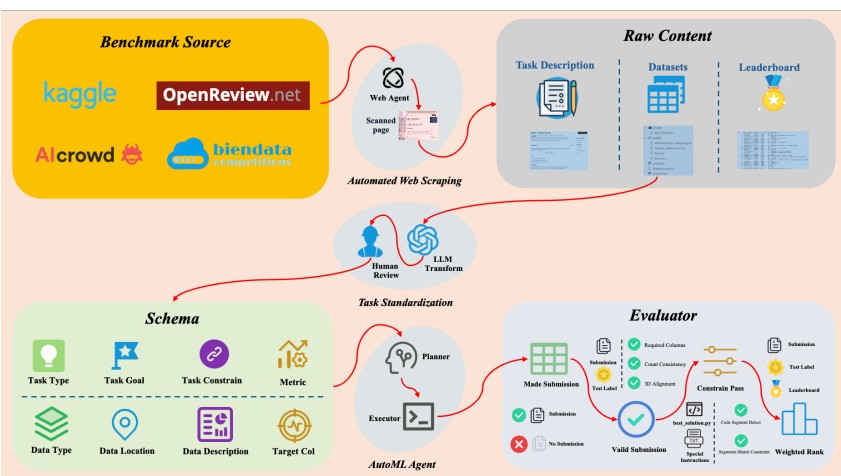

Figure 1: Overview of the system.

## 2 RELATED WORK

### 2.1 BENCHMARKS FOR END-TO-END ML TASKS

With LLMs widely adopted in data science and ML automation, several benchmark frameworks have emerged to evaluate agents' end-to-end ML task performance. **MLE-BenchChan et al. (2025)** is a benchmark built from Kaggle competitions with 75 offline challenges spanning various data types and complexity levels. It features standardized difficulty tiers (Easy/Medium/Hard) and enables fine-grained agent evaluation. **MLAgentBenchHuang et al. (2024)** focuses on LLM agents' experimental performance across 13 tasks, including image classification (e.g., CIFAR-10), NLP, and BabyLM. It supports file I/O and code execution in a unified environment. **CALMFeng et al. (2024)** targets credit risk assessment. It introduces an instruction-tuned LLM trained on over 45,000 sam-

ples from nine datasets, emphasizing reproducibility and real-world application in finance. These benchmarks standardize evaluation and foster agent development, but each has scope limitations.

## 2.2 AGENT SYSTEMS FOR ML AUTOMATION

To tackle benchmark challenges, various LLM agent systems have been proposed to automate ML pipelines from problem understanding to interpretation. **AIDEJiang et al. (2025)** (Jiang et al., 2025) and **OpenHandsWang et al. (2025)** (Wang et al., 2025) are two key systems on MLE-Bench. AIDE uses modular RL-LLM integration for Kaggle-style tasks. OpenHands, a general-purpose software agent platform, allows agents to operate OS environments in human-like ways, enabling complex workflows. These systems combine LLM reasoning with structured pipelines but still face challenges in long-term dependencies, code quality, and generalization. Future research will focus on improving autonomy, transferability, and interpretability.

## 3 METHOD

This section describes the core design and implementation of TAM-Bench, including automated pipeline, task schema, task diversity strategy, difficulty modeling, and multi-metric evaluation.

### 3.1 AUTOMATED BENCHMARK CONSTRUCTION

Traditional benchmark construction typically relies on manual processes such as competition selection, content extraction, and task filtering. Although this ensures data quality, it suffers from low efficiency and high labor costs. To address these limitations, we propose a fully automated benchmark construction method that leverages large language models (LLMs) and browser automation technologies, significantly reducing repetitive human labor while improving scalability and efficiency.To achieve this, we design a technical pipeline that automatically retrieves tasks and constructs evaluation samples from major competition platforms such as Kaggle, AIcrowd, and Biendata.Inspired by the core idea of Model Context Protocol (MCP)Hou et al. (2025)—which connects heterogeneous data sources and tools to LLMs via a unified interface—we recognize that automated benchmark construction faces similar challenges in accessing and processing multi-format, cross-platform task data. Drawing upon the design philosophy of "unified access interface + agent-driven tool collaboration," we implement an automated pipeline using the open-source WebAgent framework, `Browser-Use`Müller & Žunič (2024), to fetch and process competition tasks across platforms.

The `Browser-Use`Müller & Žunič (2024) framework enables browser control through natural language commands, simulating human web interactions without relying on platform-specific APIs, thus ensuring broad platform compatibility. We abstract web interaction behaviors into a unified agent control interface, enabling automatic extraction of task content from various competition platforms and achieving full-process automation—from task acquisition to evaluation sample generation—thus forming a closed-loop pipeline for *data access → content extraction → benchmark construction*. The core architecture consists of four hierarchical layers:

- **Agent Layer:** Based on LangChainMavroudis (2024)'s ReActYao et al. (2023) architecture, responsible for task parsing, context management, and decision generation;
- **Controller Layer:** Translates the agent's decisions into browser operation commands, serving as the bridge between the AI agent and web interfaces;
- **DOM Layer:** Parses webpage structures and converts them into AI-friendly textual representations;
- **Browser Layer:** Implements low-level browser control using Playwright, supporting multi-context management and browser instance configuration.

We enhance the controller with an `extract_markdown` tool to extract clean Markdown content from task pages by removing ads, navigation bars, etc. We then collect Task content (overview, data, evaluation) and Leaderboard data (private rankings);

A key advantage of WebAgent-based construction lies in the efficiency of task collection itself. In prior benchmarks such as MLEBench, annotators manually browsed competition pages and copied

descriptions, a process that typically required several minutes per task. For example, collecting 100 raw task descriptions could take 4–5 hours of manual effort. By contrast, our WebAgent pipeline automatically retrieves the same number of tasks within less than 1 hour, reducing the collection cost by an order of magnitude. This efficiency gain makes it feasible to continuously update TAM-Bench with new competitions, which would be prohibitively expensive under a purely manual workflow. Post-extraction filtering includes:

**Task Domain and Modality Balancing**  To address the coverage gaps in MLEBenchChan et al. (2025), TAM-Bench selectively supplements task domains and data modalities that are missing or underrepresented. We first use GPT-4o to annotate all tasks in MLEBench with standardized `field` (application domain) and `topic` (data modality) labels. Based on this analysis, we identify under-covered areas—such as e-commerce, bioinformatics, and graph-based tasks—and enrich the benchmark accordingly to ensure more balanced domain and modality representation. All candidate tasks are drawn from a larger pool of competitions and pass a strict filtering pipeline: (1) exclude those before 2023 to avoid contamination from pretraining corpora; (2) remove tasks without publicly available datasets; (3) discard tasks lacking ground-truth test labels or reproducible data splits; and (4) exclude those where the official scoring procedure cannot be replicated. This ensures that all added tasks are high-quality, reproducible, and fair, effectively extending MLEBench while maintaining rigorous standards.

**Unified schema mapping**  To support task understanding and evaluation, we convert markdown-based competition descriptions into structured schema using GPT-4o(OpenAI). The detailed schema format is provided in Appendix A.1. This structured representation serves two main purposes: First, this structured design enables tasks from different sources to be mapped into a unified format, making it easier to verify whether all essential information required for modeling is present. Second, it facilitates the extraction of key fields for downstream analysis. For example, the `metric` field can be used to support difficulty estimation and ranking, while the `special_instructions` field helps assess whether the agent adheres to specific modeling constraints during problem solving.These aspects will be discussed in detail in the subsequent sections.

**Dataset Splitting and Evaluation Consistency**  Due to the fact that competition platforms such as Kaggle do not release ground-truth labels for their test sets, direct evaluation on the original test sets is not feasible. Although delayed submission could in principle provide evaluation scores, it is constrained by strict rate limits and daily submission quotas. To address this, we reconstruct dataset splits by partitioning the official training sets into new training and test subsets, thereby generating local test labels for evaluation. To verify the reliability of this approach, we conducted a consistency check between TAM-Bench evaluation and the official Kaggle system. Specifically, we compared the scores of provided `sample_submission.csv` files and several baseline kernels submitted both to Kaggle and to our local evaluation pipeline. The results show a high degree of agreement: the Pearson correlation between local and official leaderboard scores exceeds 0.85, with a median absolute error of less than two percentile points. This indicates that our reconstructed splits provide a faithful approximation of official leaderboard standings, enabling robust offline evaluation without dependence on restricted online submission portals.

Ultimately, we construct a benchmark containing 150 tasks, referred to as the **FULL** version. Compared to existing benchmarks such as MLEBench (75 tasks), our FULL version ensures significantly broader coverage across both data modalities and application domains. In addition to traditional machine learning scenarios such as financial risk control and medical diagnosis, it also includes real-world industrial applications like e-commerce recommendation and bioinformatics, thereby capturing a wider range of practical challenges faced by intelligent agents.To address the high computational cost and token consumption associated with evaluating agents on the entire benchmark, we further propose a **Lite** version comprising 18 tasks. Despite its reduced size, the Lite version is carefully designed to achieve balanced coverage across both data modalities and task difficulty. It consists of six data modalities—Text, Tabular, Image, Audio, Graph, and MultiModal—with each modality containing three tasks categorized as easy, medium, and hard. This balance avoids the over-concentration on specific task types observed in MLEBenchChan et al. (2025) and helps prevent biased evaluation results.In addition, we introduce a **Medium** version with 54 tasks as a practical compromise between comprehensiveness and evaluation efficiency. This three-tiered benchmark structure—Lite, Medium, and Full—not only provides flexible options for evaluation under differ-

ent resource constraints but also ensures diverse, balanced, and realistic testing environments for assessing general-purpose agents. For further details, please refer to Appendix A.1.

## 3.2 AUTOMATED TASK DIFFICULTY MODELING BASED ON LEADERBOARD SIGNALS

To enable fair and scalable evaluation of AutoML systems' generalization ability across diverse tasks, we propose a novel **automated task difficulty modeling approach** based solely on leaderboard data. This method addresses two key limitations in existing AutoML benchmarks.

First, MLEBenchChan et al. (2025) relies on manually labeled difficulty levels derived from estimated task completion times. While interpretable, this strategy is inherently subjective, incurs high annotation costs, and lacks scalability. Inconsistencies across annotators may further affect labeling quality. Second, MLAgentBenchHuang et al. (2024) does not incorporate any notion of task difficulty, instead treating all tasks as equally weighted during evaluation. This ignores the intrinsic complexity differences across tasks and may distort overall performance assessment.

In real-world applications, task difficulty directly affects the required modeling capacity, feature engineering efforts, and computational budgeting strategies. Therefore, explicitly modeling task difficulty is essential for both *generalization-aware benchmarking* and *difficulty-adaptive AutoML pipeline design*.

**Difficulty Modeling via Leaderboard Structure.** Our method leverages two structural signals from competition leaderboards:The **mean score** across all participants, reflecting the average modeling capacity within the community.The **best score**, representing the performance ceiling achieved by the most capable solution. To ensure comparability across heterogeneous evaluation metrics (e.g., accuracy, loss), we apply a normalization scheme based on the metric's value range:

$$\text{NormMean} = \frac{\text{Mean Score} - \text{Min Score}}{\text{Max Score} - \text{Min Score} + \epsilon}, \tag{1}$$

$$\text{NormBest} = \frac{\text{Best Score} - \text{Min Score}}{\text{Max Score} - \text{Min Score} + \epsilon} \tag{2}$$

Here, $\epsilon$ is a small constant to prevent division by zero. For metrics with unbounded ranges, Max Score and Min Score are set to the maximum and minimum values observed on the private leaderboard. We define task difficulty differently depending on the direction of evaluation metric:

- For **higher-is-better** metrics (e.g., accuracy, AUC):

$$\text{Difficulty Score} = w_1 \cdot \text{NormMean} + w_2 \cdot \log_{10}(\text{Participants} + 1) + w_3 \cdot (1 - \text{NormBest}) \tag{3}$$

- For **lower-is-better** metrics (e.g., loss, RMSE):

$$\text{Difficulty Score} = w_1 \cdot \text{NormMean} + w_2 \cdot \log_{10}(\text{Participants} + 1) + w_3 \cdot \text{NormBest} \tag{4}$$

We empirically set the weights to $w_1 = 0.4$, $w_2 = 0.1$, and $w_3 = 0.5$. The log-scaled participant count accounts for the task's popularity or accessibility, which may influence the best achievable performance.

This formulation captures both the *community-level difficulty* (via NormMean) and the *absolute ceiling of task complexity* (via NormBest), while controlling for external factors such as exposure and engagement. To operationalize the scoring function, we categorize tasks into three difficulty levels based on their computed scores:

- **Easy**: Difficulty Score $\leq 0.6$
- **Medium**: $0.6 < \text{Score} \leq 0.85$
- **Hard**: Score $> 0.85$

**Difficulty Level Partitioning and Validation.** To validate the method's consistency, we compare our difficulty levels with those manually defined in MLEBenchChan et al. (2025) for a shared subset of tasks. Tasks labeled as "Hard" in MLEBench consistently fall within the Medium or Hard bins under our scheme, and no manually difficult task is misclassified as Easy.

Beyond this qualitative alignment, we provide a series of quantitative comparisons in Appendix A.2–A.4. Specifically, the confusion matrix (Figure 3) shows that most tasks lie near the diagonal, indicating strong agreement between TAM-Bench and MLEBench difficulty annotations. The boxplot

comparison (Figure 4) further demonstrates that difficulty levels in TAM-Bench evolve smoothly without abrupt jumps. Finally, Figure 5 highlights the 24 out of 75 competitions in MLEBench whose difficulty levels were adjusted under our scheme: only one competition transitions directly from *easy* to *hard*, while all others change gradually to an adjacent level. Taken together, these analyses provide empirical evidence that our automated difficulty modeling achieves both high consistency with expert judgment and improved granularity for scalable benchmark construction.

## 3.3 EVALUATION METRICS

Existing AutoML benchmarks often rely on overly simplistic evaluation metrics that fail to comprehensively reflect the overall modeling capabilities of AutoML systems. The table below summarizes the core metrics used by representative benchmarks:

Table 1: Evaluation metrics adopted by typical AutoML benchmarks.

| Benchmark | Primary Evaluation Metrics |
| --- | --- |
| MLEBench | *Any Medal %*: Percentage of tasks where any medal was won |
| MLAgentBench | *Competence* (success rate: +10% over baseline), *Efficiency* (latency and token cost) |
| CALM | *Model performance*, *Bias metrics* |

Although these metrics are meaningful in certain aspects, they suffer from several limitations: 1) **Task Distribution Bias**: The benchmarks lack coverage across data modalities and task difficulty levels, and the current metrics fail to account for this imbalance. 2) **Lack of Business Constraint Validation**: They do not evaluate whether the model outputs satisfy domain-specific structural or logical constraints, which may lead to "high-scoring but invalid" solutions. 3) **Weak Generalization and Stability Assessment**: Current metrics do not reflect the model's robustness across task types, modalities, and difficulty levels. 4) **Neglect of Format Compliance**: Basic format errors in submissions are often overlooked, despite their importance for real-world deployment. To address these limitations, we propose a multidimensional evaluation framework that captures the modeling capability, generalization ability, and reliability of AutoML agents.

**Weighted Average Rank**  We adopt the agent's relative leaderboard ranking percentage in real-world competitions as the core performance metric. For each task, we first compute a raw score by feeding the agent's `submission.csv` and the ground-truth `test_labels.csv` into a customized evaluation script (aligned with the competition's original evaluation protocol). This score is then converted into a rank percentage, representing the agent's percentile position on the competition leaderboard. For each task, the rank percentage is computed as:

$$\text{RankPct}_i = \frac{\text{rank}_i}{\text{total participants}_i}$$

To mitigate bias caused by overrepresentation of tasks from certain modalities, we apply modality-aware weighting:

$$\text{WeightedRank} = \frac{\sum_{i=1}^{N} \frac{1}{f(m_i)} \cdot \text{RankPct}_i}{\sum_{i=1}^{N} \frac{1}{f(m_i)}}$$

where:

- $N$ is the total number of evaluated tasks;
- $m_i$ denotes the data modality of the $i$-th task;
- $f(m_i)$ is the frequency of modality $m_i$ in evaluation set;
- $\text{RankPct}_i$ is the agent's percentile rank on task $i$.

This modality-weighted approach reduces evaluation bias due to task imbalance and highlights an agent's **generalization performance across modalities**. We also report grouped average rank percentages across task difficulty levels (Easy / Medium / Hard) and modalities to provide deeper insights into the agent's stability and limitations.

**Constraint Pass Rate**   To assess whether the agent adheres to the `special_instructions` field of each task, we employ an LLM-as-a-Judge evaluation method. Specifically:

- We prompt an LLM with the special instructions and the final submitted code to determine if all required constraints are satisfied.

- Each constraint is marked as `Passed`/`Failed`, and the overall pass rate is computed as:

$$\text{ConstraintPass} = \frac{\text{Number of Passed Constraints}}{\text{Total Constraints}}$$

Importantly, we evaluate constraints based solely on the **final submitted code**, rather than interme­diate logs or code fragments, for the following reasons: 1) **Finality and Executability**: Intermediate code may be incomplete or overwritten later. Only the final submission reflects the agent's intended solution. 2) **Semantic Completeness**: Logs and partial outputs are often ambiguous or missing key information, making them unsuitable for accurate constraint checking. 3) **Outcome-Oriented Eval­uation**: We prioritize whether the final deliverable satisfies task requirements, rather than whether the agent attempted to follow constraints.

This metric helps prevent *reward hacking*, where agents achieve superficially high scores without complying with task requirements. By evaluating code-level compliance, we improve the **robust­ness and reliability** of the entire evaluation system. For specific details regarding the prompt, please refer to Appendix A.2.

**Format Compliance**   Format compliance is a basic yet critical requirement for any AutoML sys­tem. We evaluate it on two levels: 1) **Made Submission**: The proportion of tasks for which the agent successfully generates a `submission.csv` file. 2) **Valid Submission**: The proportion of tasks where the generated file matches the structure of the provided `test_labels.csv`, with no missing columns, misaligned formats, or structural errors. These two metrics capture the agent's capability to generate submission outputs and follow required format specifications—both essential for practical deployment. A reliable AutoML agent must demonstrate not only modeling ability but also robustness and correctness in file handling and formatting.

In summary, our evaluation framework comprises three dimensions—**performance ranking**, **con­straint compliance**, and **format validity**. Together, they form a comprehensive, multi-faceted, and interpretable system for measuring the true capabilities of AutoML agents, promoting more reliable and accountable AutoML development.

## 4 EXPERIMENT

### 4.1 EXPERIMENTAL SETUP AND EVALUATION PROTOCOL

In our experiments, we evaluated two open-source AutoML Agent frameworks: **AIDEJiang et al. (2025)** and **OpenHandsWang et al. (2025)**, each paired with two different base language mod­els: **GPT-4.1OpenAI (2024)** (closed-source) and **DeepSeek-V3DeepSeek-AI et al. (2025)** (open-source). The rationale for model selection is as follows:

For open-source models, we initially experimented with the Qwen family (e.g., Qwen3-Coder, Qwen-Max), but encountered JSON parsing errors during execution, which significantly impacted task stability and completion rates. Considering both stability and overall performance, we ulti­mately selected DeepSeek-v3 to represent open-source models. For closed-source models, since AutoML tasks involve not only code generation and debugging but also require strong task under­standing and planning capabilities, we chose GPT-4.1, which has demonstrated top performance across multiple benchmarks, as the representative closed-source model.

To ensure manageable computational costs, we set AIDEJiang et al. (2025)'s `agent_steps` to 10 for all tests. When switching base models, we replaced the internal code model, feedback model, and report model with their corresponding variants based on the selected base model to ensure fair eval­uation. All other parameters were kept at their default values. For OpenHandsWang et al. (2025), apart from the base model replacement, all other configurations also remained at default.

Considering the long evaluation cycle of individual benchmark tasks, we used the **Lite version Benchmark** in this paper to assess the overall performance of Agents. This benchmark comprises 18 public competition tasks. All experiments were conducted within Docker containers running

Ubuntu 20.04, each containing the task description and dataset. Each Agent was allowed a maximum runtime of **8 hours**, and access to a single machine with **16 vCPUs, 60 GiB RAM, and an NVIDIA A10 GPU**. The Agent's execution process includes reading task documentation, performing data analysis and model tuning, and finally generating two outputs: a prediction file named `submission.csv` and a solution script named `best_solution.py`. These two files are then fed into our evaluation system, which computes four key metrics: Made Submission, Valid Submission, Constraint Pass Rate, and Average Rank.

## 4.2 DISCUSSION

To comprehensively evaluate AutoML agents, we first examine their **modeling compliance** in practical tasks across the following key dimensions:

- **Made Submission**: Number of tasks where the agent successfully generated a `submission.csv` file.
- **Valid Submission**: Number of tasks where the `submission.csv` file format was correct and matched `test_labels.csv`.
- **Average Constraint Pass**: Average pass rate of modeling constraints (verified by GPT-4o).
- **Average Rank**: Average leaderboard percentile across all tasks.

Table 2: Submission statistics, constraint pass rates, and average rank by framework and model.

| Model | Made Submission (%) | Valid Submission (%) | Average Constraint Pass (%) | Average Rank (%) |
|---|---|---|---|---|
| **AIDE** | | | | |
| GPT-4.1 | 72 | 56 | 88.2 | 87 |
| DeepSeek-V3 | 39 | 28 | 90.4 | 86 |
| **OpenHands** | | | | |
| GPT-4.1 | 78 | 56 | 84.3 | 88 |
| DeepSeek-V3 | 56 | 28 | 86.9 | 91 |

The results reveal that while most agents are capable of producing outputs in the majority of tasks, a significant fraction of the generated `submission.csv` files fail to meet the required format specifications—particularly for DeepSeek-V3-based agents, likely due to less robust code generation. In contrast, agents using GPT-4.1OpenAI (2024) show higher rates of valid submissions, reflecting better adherence to formatting and output conventions.

In terms of constraint compliance, most agents maintain high pass rates (close to 100%) on general tasks. However, tasks with stringent requirements (e.g., "must use MFCC features" or "must perform time-series decomposition") are often violated. AIDEJiang et al. (2025) supported by GPT-4.1OpenAI (2024) consistently achieves higher compliance, underscoring the importance of model capability in understanding and implementing complex constraints.

Interestingly, when considering the **Average Rank**, DeepSeek-V3DeepSeek-AI et al. (2025) agents slightly outperform GPT-4.1OpenAI (2024) variants. For example, AIDE + DeepSeek-V3 achieves an average rank percentile of 86%, compared to 87% for its GPT-4.1 counterpart. Similarly, OpenHands + DeepSeek-V3 ranks at 91%, slightly worse than OpenHands + GPT-4.1 (88%). However, the differences are small, and given the much lower submission and validity rates of DeepSeek-V3, this ranking advantage may be biased toward the subset of tasks where successful output was achieved.

Overall, these results suggest that while DeepSeek-V3DeepSeek-AI et al. (2025) has the potential to produce highly competitive solutions when it works, GPT-4.1OpenAI (2024) demonstrates significantly better reliability, making it a stronger choice for end-to-end AutoML agent deployment.

**Performance Across Task Difficulties** Beyond modeling compliance, we further analyze the agents' performance under varying **task difficulties** and **data modalities**. All agents perform more stably on **EASY** tasks, with relatively lower average ranking percentiles. However, as difficulty

Table 3: Average rank scores across difficulty levels (lower is better).

| Difficulty | AIDE | | OpenHands | | Average Rank |
|---|---|---|---|---|---|
| | GPT-4.1 | DeepSeek-V3 | GPT-4.1 | DeepSeek-V3 | |
| **EASY** | 76 | 88 | 83 | 75 | 80.5 |
| **MEDIUM** | 89 | 89 | 90 | 98 | 91.5 |
| **HARD** | 95 | 82 | 94 | 100 | 92.75 |

increases to **MEDIUM** and **HARD**, percentile values rise sharply, often exceeding 80%—and even reaching 100% in some cases. This indicates persistent performance bottlenecks in handling complex tasks, likely due to inadequate feature engineering or failure to meet critical constraints. From a base model perspective, GPT-4.1-based agents generally outperform their DeepSeek-V3 counterparts, especially on MEDIUM and HARD tasks, demonstrating stronger language understanding and global reasoning capabilities. However, DeepSeek-V3DeepSeek-AI et al. (2025) exhibits competitive performance in some EASY tasks and in structure-heavy modalities such as Tabular and Text, highlighting its robustness and execution-level fault tolerance.

Table 4: Average rank scores across Category (lower is better).

| Category | AIDE | | OpenHands | | Average Rank |
|---|---|---|---|---|---|
| | GPT-4.1 | DeepSeek-V3 | GPT-4.1 | DeepSeek-V3 | |
| **Tabular** | 82 | 67 | 71 | 93 | 78.25 |
| **Text** | 71 | 58 | 95 | 76 | 75 |
| **Image** | 92 | 92 | 90 | 90 | 91 |
| **Audio** | 89 | 100 | 89 | 86 | 91 |
| **Graph** | 100 | 100 | 100 | 100 | 100 |
| **Multi-Modal** | 86 | 100 | 85 | 100 | 92.75 |

**Modality-Specific Trends** In terms of data modality, agents tend to achieve better results on **Tabular** and **Text** tasks, suggesting relatively mature capabilities in handling conventional data types. By contrast, performance on **Image** and **Audio** tasks remains weak across all agents, pointing to significant limitations in modeling perceptual modalities. These results suggest that LLM-centric architectures lack the specialized feature extraction and inductive biases needed for high-dimensional visual and acoustic data. Especially in **Graph** and **Multi-Modal** tasks, most agents fall into the 100% percentile range, revealing a critical gap in handling structured and heterogeneous information.

**Case Highlight: Breakthrough on Tabular Hard Task** A particularly noteworthy result comes from the **OpenHands + DeepSeek-V3** combination, which achieved first place on the challenging `stanford-covid-vaccine` task with an MCRMSE score of **0.30**, surpassing the best human team's score of 0.34198. This demonstrates that certain agent-model combinations, when well-aligned with task characteristics, can outperform traditional human-designed pipelines. It also highlights the potential of LLM-driven agents to navigate complex solution spaces via advanced code generation, model tuning, and search strategies.

## 5 LIMITATIONS

Our current evaluation is limited to the **Lite** version of TAM-Bench. In future work, we plan to progressively extend experiments to the **Medium** and **Full** versions, incorporating a broader set of tasks and evaluating more recent AutoML systems such as *AutoMindOu et al. (2025)*, *ML-AgentLiu et al. (2025b)*, and *ML-MasterLiu et al. (2025a)*.

Secondly, the results presented in this paper are based on single-run evaluations (`pass@1`). We will repeat all experiments and report averaged results along with confidence intervals (e.g., $97 \pm 2.7$) to ensure more statistically robust comparisons.

Finally, we are actively expanding the size of the **Full** version of TAM-Bench. As the benchmark grows, we aim to maintain a relatively balanced distribution of tasks across different data modalities and application types, aligning with our goal of comprehensive and fair evaluation.

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

## A APPENDIX

### A.1 USE OF LLMS

We used Large Language Models (LLMs) to aid and polish the writing of this paper. This usage was limited to language refinement and did not involve the generation of core ideas, methodology, or experimental results.

### A.2 DISTRIBUTION OF TASK DIFFICULTY BY MODALITY

The prompt of LLM-as-a-Judge evaluation is as follows:

```
You are a code review expert responsible for determining
whether a piece of model training code complies with
specific modeling constraints.
Please evaluate the following constraint based on the code
below:
- Constraint {idx}:  {constraint}
Code:
{code_content}
Your response must follow this format exactly:
Answer:  [Yes/No]
Reason:  [Brief and specific justification explaining why
the code complies with the constraint or lacks sufficient
information to verify compliance]
Important:
If the code does not include relevant evidence,
conservatively answer "No".
Do not include any additional explanations, comments, or
formatting beyond the required two lines.
```

### A.3 DISTRIBUTION OF TASK DIFFICULTY BY MODALITY

Figure 2 presents the difficulty distribution across different data modalities for the 150 tasks in the full version of TAM-Bench. As shown, TAM-Bench covers all three difficulty levels—*easy*, *medium*, and *hard*—across six data modalities, avoiding the over-representation issues observed in MLE-Bench.

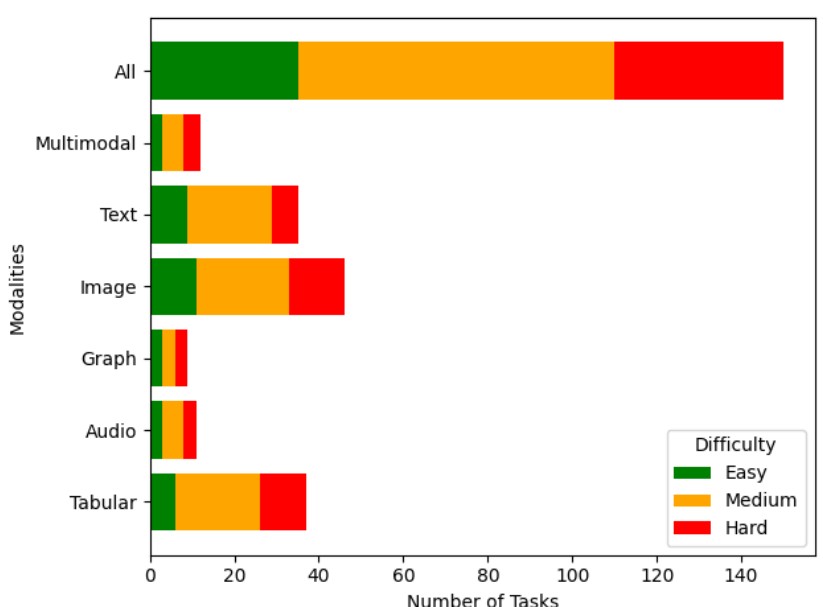

Figure 2: Distribution of Task Difficulty by Modality

### A.4    DIFFICULTY CONFUSION MATRIX: TAM-BENCH VS. MLE-BENCH

Figure 3 displays the confusion matrix comparing the difficulty grading of TAM-Bench and MLE-Bench. The majority of tasks lie near the diagonal, indicating a high degree of alignment between the two benchmarks. Notably, no task was reassigned across two difficulty levels (e.g., from easy to hard or vice versa), except for one instance where a hard task in MLE-Bench was downgraded to easy in TAM-Bench, which stands out as an outlier.

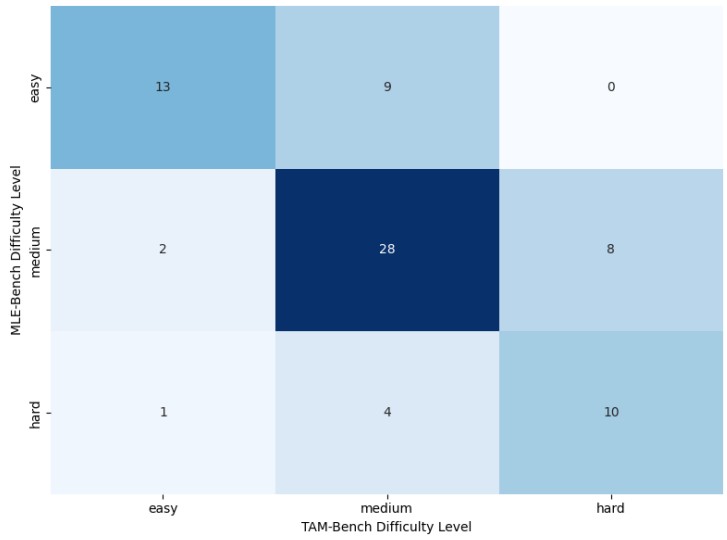

Figure 3: Difficulty Confusion Matrix: TAM-Bench vs. MLE-Bench

## A.5    COMPARISON OF DIFFICULTY LEVELS BETWEEN MLE-BENCH AND TAM-BENCH

Figure 4 shows the difficulty reclassification of MLE-Bench tasks under the TAM-Bench framework. A significant proportion of originally labeled easy tasks (41%) were reassigned to medium, suggesting that TAM-Bench adopts a stricter criterion for simplicity. Conversely, about one-third of hard-labeled tasks in MLE-Bench were downgraded—27% to medium and 7% to easy—indicating potential overestimation of difficulty in the original benchmark. While the majority of tasks retain their general difficulty level (e.g., 74% of medium tasks remain unchanged), the transitions reveal a more refined calibration in TAM-Bench, particularly in distinguishing between medium and hard problems. These adjustments highlight the importance of standardized difficulty annotation and suggest that TAM-Bench provides a more balanced and empirically grounded difficulty distribution.

Figure 4: Percentage Distribution of MLE-Bench Tasks in TAM-Bench

## A.6    24 COMPETITIONS IN MLE-BENCH THAT CHANGED IN DIFFICULTY

Table 5 shows the mapping of 24 out of 75 MLE-Bench competitions whose difficulty levels were adjusted in TAM-Bench.Only 32% of the competitions experienced a change in difficulty level after applying our classification criteria, while the remaining 68% maintained the same difficulty level as in *MLE-Bench*.

## A.7    CASE OF THE STRUCTURED SCHEMA

Figure 5 illustrates the standardized schema proposed in this work. The `special_instruction` field is particularly used for evaluating constraint compliance.

## A.8    EXAMPLE OF CONSTRAINT PASS EVALUATION

Figure 6 shows a sample case used to evaluate the *Constraint Pass Rate*. Based on the `best_solution.py` and the `special_instruction` field, we apply an LLM-as-judge method. Green boxes in the figure highlight code segments that comply with constraints, while red boxes indicate violations. The resulting Constraint Pass Rate for this case is 66.7%.

## A.9    ORIGINAL EXPERIMENTAL RESULTS ON TAM-BENCH LITE VERSION

Table 6 shows the original experimental results of AIDE with GPT-4.1 on the TAM-Bench Lite Version.
Table 7 shows the original experimental results of OpenHands with GPT-4.1 on the TAM-Bench Lite Version.

Table 5: Difficulty Comparison of 24 Competitions in MLEBench with Changed Difficulty Levels

| Competition | MLE-Bench Difficulty | TAM-Bench Difficulty |
|---|---|---|
| rsna-2022-cervical-spine-fracture-detection | Hard | Medium |
| predict-volcanic-eruptions-ingv-oe | Hard | Medium |
| nfl-player-contact-detection | Hard | Medium |
| hms-harmful-brain-activity-classification | Hard | Medium |
| bms-molecular-translation | Hard | Easy |
| whale-categorization-playground | Medium | Hard |
| tweet-sentiment-extraction | Medium | Hard |
| tensorflow2-question-answering | Medium | Hard |
| osic-pulmonary-fibrosis-progression | Medium | Hard |
| multi-modal-gesture-recognition | Medium | Easy |
| hubmap-kidney-segmentation | Medium | Hard |
| h-and-m-personalized-fashion-recommendations | Medium | Hard |
| google-quest-challenge | Medium | Hard |
| cdiscount-image-classification-challenge | Medium | Hard |
| billion-word-imputation | Medium | Easy |
| tabular-playground-series-dec-2021 | Easy | Medium |
| siim-isic-melanoma-classification | Easy | Medium |
| ranzcr-clip-catheter-line-classification | Easy | Medium |
| random-acts-of-pizza | Easy | Medium |
| plant-pathology-2020-fgvc7 | Easy | Medium |
| nomad2018-predict-transparent-conductors | Easy | Medium |
| jigsaw-toxic-comment-classification-challenge | Easy | Medium |
| histopathologic-cancer-detection | Easy | Medium |
| aptos2019-blindness-detection | Easy | Medium |

```
{
  "task_type": "classification",
  "goal_description": "Build an algorithm that predicts the correct label for simple spoken commands from audio clips.",
  "metric": {
    "metric_name": "Multiclass Accuracy",
    "metric_formula": ""
  },
  "target_col": "label",
  "data_information": {
    "data_type": "Audio",
    "train": {
      "data_location": "train.7z",
      "data_description": "Contains a few informational files and a folder of audio files. The audio folder contains subfolders with 1-second clips of voice commands, with the folder name
        being the label of the audio clip. Labels include `yes`, `no`, `up`, `down`, `left`, `right`, `on`, `off`, `stop`, `go`, `silence`, and `unknown`. The `_background_noise_` folder
        contains longer clips of `silence` that can be used for training. Audio files are not uniquely named across labels but are unique when including the label folder. Files have
        inconsistent properties such as length. Features to extract could include MFCCs, spectrograms, or raw audio signals."
    },
    "test": {
      "data_location": "test.7z",
      "data_description": "Contains an audio folder with 150,000+ files in the format `clip_*.wav`. The task is to predict the correct label for each file. Not all files are evaluated for
        the leaderboard score. Test data may contain unseen subjects and should be processed accordingly."
    },
    "inference": {
      "data_location": "",
      "data_description": ""
    }
  },
  "output_format": "fname,label\nclip_000044442.wav,silence\nclip_0000adecb.wav,left\nclip_0000d4322.wav,unknown\netc.",
  "special_instructions": "1.The `unknown` label should be used for any command that is not one of the first 10 labels (`yes`, `no`, `up`, `down`, `left`, `right`, `on`, `off`, `stop`,
    `go`) or that is not `silence`.2. Use the `_background_noise_` folder to generate additional silence samples for training if needed. 3. Must-use features: Consider extracting acoustic
    features such as MFCCs (Mel-Frequency Cepstral Coefficients) or spectrograms for model input."
}
```

Figure 5: The structured schema of tensorflow-speech-recognition-challenge_result task

756
757
758
759
760
761
762
763
764
765
766
767
768
769
770
771
772
773
774
775
776
777
778
779
780
781
782
783
784
785
786
787
788
789
790
791
792
793
794
795
796
797
798
799
800
801
802
803
804
805
806
807
808
809

**Tensorflow-speech-recognition-challenge Constraints**

1. The unknown label should be used for any command that is not one of the first 10 labels (yes, no, up, down, left, right, on, off, stop, go) or that is not silence.

2. Use the _background_noise_ folder to generate additional silence samples for training if needed.

3. Must-use features: Consider extracting acoustic features such as MFCCs (Mel-Frequency Cepstral Coefficients) or spectrograms for model input.

**Openhand Code**

```python
import os
import numpy as np
import pandas as pd
import scipy.io.wavfile as wav
from glob import glob
from tqdm import tqdm
from sklearn.ensemble import RandomForestClassifier
from sklearn.model_selection import train_test_split
from sklearn.metrics import accuracy_score

# Main labels for the challenge
target_labels = ['yes', 'no', 'up', 'down', 'left', 'right', 'on', 'off', 'stop', 'go']
all_labels = target_labels + ['silence', 'unknown']

def extract_features(signal, rate):
    # Simple features: mean, std, max, min, energy, zero-crossing rate, first 10 FFT bins
    signal = signal.astype(np.float32)
    mean = np.mean(signal)
    std = np.std(signal)
    maxv = np.max(signal)
    minv = np.min(signal)
    energy = np.sum(signal ** 2) / len(signal)
    zcr = ((signal[:-1] * signal[1:]) < 0).sum() / len(signal)
    fft = np.abs(np.fft.rfft(signal, n=512))[:10]
    return np.array([mean, std, maxv, minv, energy, zcr] + list(fft))
```

*Constraint 3*

```python
def get_silence_samples(bg_dir, sample_len=16000, n_samples=200):
    # Chop background noise into 1s samples
    X = []
    for fname in glob(os.path.join(bg_dir, '*.wav')):
        rate, data = wav.read(fname)
        for i in range(0, len(data) - sample_len, sample_len):
            chunk = data[i:i+sample_len]
            if len(chunk) == sample_len:
                X.append(chunk)
            if len(X) >= n_samples:
                break
        if len(X) >= n_samples:
            break
    return X
```

*Constraint 2*

```python
def load_train_data(audio_dir, bg_dir):
    X, y = [], []
    for label in os.listdir(audio_dir):
        label_dir = os.path.join(audio_dir, label)
        if not os.path.isdir(label_dir):
            continue
        if label == '_background_noise_':
            continue
        files = glob(os.path.join(label_dir, '*.wav'))
```

```
    for f in files:
        rate, data = wav.read(f)
        if data.shape[0] != 16000:
            continue  # skip non-1s clips
        X.append(extract_features(data, rate))
        if label in target_labels:          Constraint 1
            y.append(label)
        else:
            y.append('unknown')
    # Add silence samples
    for chunk in get_silence_samples(bg_dir):
        X.append(extract_features(chunk, 16000))
        y.append('silence')
    return np.array(X), np.array(y)

def load_test_data(test_dir):
    X, fnames = [], []
    files = sorted(glob(os.path.join(test_dir, '*.wav')))
    for f in tqdm(files, desc='Extracting test features'):
        rate, data = wav.read(f)
        if data.shape[0] != 16000:
            # pad or trim
            if data.shape[0] > 16000:
                data = data[:16000]
            else:
                data = np.pad(data, (0, 16000 - data.shape[0]), 'constant')
        X.append(extract_features(data, rate))
        fnames.append(os.path.basename(f))
    return np.array(X), fnames

def main():
    train_audio_dir = 'tensorflow-speech-recognition-challenge/train/audio'
    bg_dir = os.path.join(train_audio_dir, '_background_noise_')
    test_audio_dir = 'tensorflow-speech-recognition-challenge/test/audio'
    print('Loading train data...')
    X, y = load_train_data(train_audio_dir, bg_dir)
    print('Train data shape:', X.shape, y.shape)
    X_train, X_val, y_train, y_val = train_test_split(X, y, test_size=0.1, random_state=42, stratify=y)
    clf = RandomForestClassifier(n_estimators=100, n_jobs=-1, random_state=42)
    clf.fit(X_train, y_train)
    y_pred = clf.predict(X_val)
    print('Validation accuracy:', accuracy_score(y_val, y_pred))
    print('Loading test data...')
    X_test, fnames = load_test_data(test_audio_dir)
    print('Predicting test...')
    y_test_pred = clf.predict(X_test)
    sub = pd.DataFrame({'fname': fnames, 'label': y_test_pred})
    sub.to_csv('submission.csv', index=False)
    print('submission.csv generated.')

if __name__ == '__main__':
    main()
```

**Constrain Judge Result**

Constraint 1 ✅
Constraint 2 ✅
Constraint 3 ❌

**Reason**

The extract_features function only computes basic statistical features (mean, std, max, min), energy, zero-crossing rate, and the first 10 FFT bins. However, it does not extract any standard acoustic features such as MFCCs, mel-spectrograms, or log-mel features.

Figure 6: example of constraint pass

Table 8 shows the original experimental results of AIDE with Deepseek-v3 on the TAM-Bench Lite Version.

Table 9 shows the original experimental results of Openhands with Deepseek-v3 on the TAM-Bench Lite Version.

Table 6: Performance of AIDE with GPT-4.1 across modalities and difficulties, including raw scores and rank percentages. A dash ("-") indicates that no `submission.csv` was generated or the generated file failed to meet the required format for evaluation.

| Task Name | Modality | Difficulty | Raw Score | Rank Percentage |
|---|---|---|---|---|
| new-york-city-taxi-fare-prediction | Tabular | Easy | 11.462946 | 0.98 |
| Binary Prediction of Poisonous Mushrooms | Tabular | Medium | 0.980426 | 0.67 |
| stanford-covid-vaccine | Tabular | Hard | 0.389836 | 0.80 |
| text-normalization-challenge-english-language | Text | Easy | 0.9906 | 0.27 |
| lmsys-chatbot-arena | Text | Medium | 1.133694 | 0.96 |
| eedi-mining-misconceptions-in-mathematics | Text | Hard | 0.120925 | 0.90 |
| denoising-dirty-documents | Image | Easy | 0.226320 | 0.89 |
| statoil-iceberg-classifier-challenge | Image | Medium | 0.444734 | 0.87 |
| 3d-object-detection-for-autonomous-vehicles | Image | Hard | - | - |
| mlsp-2013-birds | Audio | Easy | 0.807922 | 0.68 |
| tensorflow-speech-recognition-challenge | Audio | Medium | - | - |
| Cornell Birdcall Identification | Audio | Hard | - | - |
| IND | Graph | Easy | - | - |
| PST | Graph | Medium | - | - |
| AQA | Graph | Hard | - | - |
| multi-modal-gesture-recognition | MultiModal | Easy | 0.78394 | 0.75 |
| NextProductPrediction | MultiModal | Medium | 0.0025 | 0.83 |
| planttraits2024 | MultiModal | Hard | - | - |

## A.10 ORIGINAL EXPERIMENTAL RESULTS ON TAM-BENCH LITE VERSION

Table 10 lists the 18 competitions included in the Lite version of TAM-Bench, along with their associated data modalities and difficulty levels.

Table 11 lists the 54 competitions included in the Medium version of TAM-Bench, along with their associated data modalities and difficulty levels.

Table 12 lists the 18 competitions included in the Full version of TAM-Bench, along with their associated data modalities and difficulty levels.

Table 7: Performance of OpenHandsWang et al. (2025) with GPT-4.1 across modalities and difficulties, including raw scores and rank percentages. A dash ("-") indicates that no `submission.csv` was generated or the generated file failed to meet the required format for evaluation.

| Task Name | Modality | Difficulty | Raw Score | Rank Percentage |
|---|---|---|---|---|
| new-york-city-taxi-fare-prediction | Tabular | Easy | 4.329638 | 0.71 |
| Binary Prediction of Poisonous Mushrooms | Tabular | Medium | 0.982892 | 0.53 |
| stanford-covid-vaccine | Tabular | Hard | 0.427988 | 0.90 |
| text-normalization-challenge-english-language | Text | Easy | 0.9606 | 0.95 |
| lmsys-chatbot-arena | Text | Medium | - | - |
| eedi-mining-misconceptions-in-mathematics | Text | Hard | 0.000560 | 0.99 |
| denoising-dirty-documents | Image | Easy | 0.152417 | 0.83 |
| statoil-iceberg-classifier-challenge | Image | Medium | 0.420190 | 0.86 |
| 3d-object-detection-for-autonomous-vehicles | Image | Hard | - | - |
| mlsp-2013-birds | Audio | Easy | 0.816988 | 0.67 |
| tensorflow-speech-recognition-challenge | Audio | Medium | 0.007415 | 1.0 |
| Cornell Birdcall Identification | Audio | Hard | - | - |
| IND | Graph | Easy | - | - |
| PST | Graph | Medium | - | - |
| AQA | Graph | Hard | - | - |
| multi-modal-gesture-recognition | MultiModal | Easy | 0.82503 | 0.80 |
| NextProductPrediction | MultiModal | Medium | 0 | 1.0 |
| planttraits2024 | MultiModal | Hard | 0.0459 | 0.75 |

Table 8: Performance of AIDE with DeepSeek-V3 across modalities and difficulties, including raw scores and rank percentages (lower is better). A dash ("-") indicates that no `submission.csv` was generated or the generated file failed to meet the required format for evaluation.

| Task Name | Modality | Difficulty | Raw Score | Rank Percentage |
|---|---|---|---|---|
| new-york-city-taxi-fare-prediction | Tabular | Easy | - | - |
| Binary Prediction of Poisonous Mushrooms | Tabular | Medium | - | - |
| stanford-covid-vaccine | Tabular | Hard | 0.309915 | **0.0006** |
| text-normalization-challenge-english-language | Text | Easy | 0.9906 | 0.27 |
| lmsys-chatbot-arena | Text | Medium | 1.057080 | 0.55 |
| eedi-mining-misconceptions-in-mathematics | Text | Hard | 0.079931 | 0.92 |
| denoising-dirty-documents | Image | Easy | - | - |
| statoil-iceberg-classifier-challenge | Image | Medium | 0.294361 | 0.77 |
| 3d-object-detection-for-autonomous-vehicles | Image | Hard | - | - |
| mlsp-2013-birds | Audio | Easy | - | - |
| tensorflow-speech-recognition-challenge | Audio | Medium | - | - |
| Cornell Birdcall Identification | Audio | Hard | - | - |
| IND | Graph | Easy | - | - |
| PST | Graph | Medium | - | - |
| AQA | Graph | Hard | - | - |
| multi-modal-gesture-recognition | MultiModal | Easy | - | - |
| NextProductPrediction | MultiModal | Medium | - | - |
| planttraits2024 | MultiModal | Hard | - | - |

Table 9: Performance of OpenHands with DeepSeek-V3 across modalities and difficulties, including raw scores and rank percentages (lower is better). A dash ("-") indicates that no `submission.csv` was generated or the generated file failed to meet the required format for evaluation.

| Task Name | Modality | Difficulty | Raw Score | Rank Percentage |
|---|---|---|---|---|
| new-york-city-taxi-fare-prediction | Tabular | Easy | 5.303500 | 0.80 |
| Binary Prediction of Poisonous Mushrooms | Tabular | Medium | - | - |
| stanford-covid-vaccine | Tabular | Hard | - | - |
| text-normalization-challenge-english-language | Text | Easy | 0.9906 | 0.27 |
| lmsys-chatbot-arena | Text | Medium | - | - |
| eedi-mining-misconceptions-in-mathematics | Text | Hard | - | - |
| denoising-dirty-documents | Image | Easy | 0.151466 | 0.83 |
| statoil-iceberg-classifier-challenge | Image | Medium | 0.403900 | 0.86 |
| 3d-object-detection-for-autonomous-vehicles | Image | Hard | - | - |
| mlsp-2013-birds | Audio | Easy | 0.847697 | 0.58 |
| tensorflow-speech-recognition-challenge | Audio | Medium | - | - |
| Cornell Birdcall Identification | Audio | Hard | - | - |
| IND | Graph | Easy | - | - |
| PST | Graph | Medium | - | - |
| AQA | Graph | Hard | - | - |
| multi-modal-gesture-recognition | MultiModal | Easy | - | - |
| NextProductPrediction | MultiModal | Medium | 0 | 1.0 |
| planttraits2024 | MultiModal | Hard | - | - |

Table 10: Overview of 18 Competitions(Lite Version), Data Modalities, and Task Complexity.

| Competition | Data Modality | Task Difficulty |
|---|---|---|
| new-york-city-taxi-fare-prediction | Tabular | easy |
| Binary Prediction of Poisonous Mushrooms | Tabular | medium |
| stanford-covid-vaccine | Tabular | hard |
| text-normalization-challenge-english-language | Text | easy |
| lmsys-chatbot-arena | Text | medium |
| eedi-mining-misconceptions-in-mathematics | Text | hard |
| denoising-dirty-documents | Image | easy |
| statoil-iceberg-classifier-challenge | Image | medium |
| 3d-object-detection-for-autonomous-vehicles | Image | hard |
| mlsp-2013-birds | Audio | easy |
| tensorflow-speech-recognition-challenge | Audio | medium |
| Cornell Birdcall Identification | Audio | hard |
| WhoIsWho-IND | Graph | easy |
| PST | Graph | medium |
| AQA | Graph | hard |
| multi-modal-gesture-recognition | Multimodal | easy |
| NextProductPrediction | Multimodal | medium |
| planttraits2024 | Multimodal | hard |

Table 11: Overview of 18 Competitions(Lite Version), Data Modalities, and Task Complexity.

| Competition | Data Modality | Task Difficulty |
| --- | --- | --- |
| um-game-playing-strength-of-mcts-variants | Tabular | easy |
| march-machine-learning-mania-2024 | Tabular | easy |
| new-york-city-taxi-fare-prediction | Tabular | easy |
| Binary Prediction of Poisonous Mushrooms | Tabular | medium |
| hms-harmful-brain-activity-classification | Tabular | medium |
| open-problems-single-cell-perturbations | Tabular | medium |
| home-credit-credit-risk-model-stability | Tabular | hard |
| stanford-covid-vaccine | Tabular | hard |
| equity-post-HCT-survival-predictions | Tabular | hard |
| Multi-Lingual Abilities | Text | easy |
| text-normalization-challenge-english-language | Text | easy |
| Shopping Knowledge Reasoning | Text | easy |
| santa-2024 | Text | medium |
| lmsys-chatbot-arena | Text | medium |
| movie-review-sentiment-analysis-kernels-only | Text | medium |
| eedi-mining-misconceptions-in-mathematics | Text | hard |
| google-quest-challenge | Text | hard |
| llms-you-cant-please-them-all | Text | hard |
| dogs-vs-cats-redux-kernels-edition | Image | easy |
| denoising-dirty-documents | Image | easy |
| aerial-cactus-identification | Image | easy |
| siim-isic-melanoma-classification | Image | medium |
| statoil-iceberg-classifier-challenge | Image | medium |
| aptos2019-blindness-detection | Image | medium |
| 3d-object-detection-for-autonomous-vehicles | Image | hard |
| iwildcam-2019-fgvc6 | Image | hard |
| image-matching-challenge-2024 | Image | hard |
| WhoIsWho-IND | Graph | easy |
| Graph based Recommendation | Graph | easy |
| PMLDL | Graph | easy |
| PST | Graph | medium |
| OGB-LSC-MAG240M | Graph | medium |
| predict-ai-model-runtime | Graph | medium |
| OGB-LSC-PCQM4M | Graph | hard |
| AQA | Graph | hard |
| OGB-LSC-WikiKG90M | Graph | hard |
| the-icml-2013-whale-challenge-right-whale-redux | Audio | easy |
| Marinexplore and Cornell Whale Detection Challenge | Audio | easy |
| the-icml-2013-whale-challenge-right-whale-redux | Audio | easy |
| mlsp-2013-birds | Audio | easy |
| tensorflow-speech-recognition-challenge | Audio | medium |
| Rainforest Connection Species Audio Detection | Audio | medium |
| bengaliai-speech | Audio | medium |
| Cornell Birdcall Identification | Audio | hard |
| BirdCLEF 2021 - Birdcall Identification | Audio | hard |
| birdclef-2024 | Audio | hard |
| User Behavior Alignment | Multimodal | easy |
| All-Around | Multimodal | easy |
| multi-modal-gesture-recognition | Multimodal | easy |
| NextProductPrediction | Multimodal | medium |
| nfl-player-contact-detection | Multimodal | medium |
| random-acts-of-pizza | Multimodal | medium |
| leash-BELKA | Multimodal | hard |
| planttraits2024 | Multimodal | hard |
| drawing-with-llms | Multimodal | hard |

Table 12: Overview of 150 Competitions(FULL Version), Data Modalities, and Task Complexity.

| Competition | Data Modality | Task Difficulty |
|---|---|---|
| linking-writing-processes-to-writing-quality | Tabular | easy |
| tabular-playground-series-may-2022 | Tabular | easy |
| um-game-playing-strength-of-mcts-variants | Tabular | easy |
| march-machine-learning-mania-2024 | Tabular | easy |
| new-york-city-taxi-fare-prediction | Tabular | easy |
| leap-atmospheric-physics-ai-climsim | Tabular | easy |
| tabular-playground-series-dec-2021 | Tabular | medium |
| icecube-neutrinos-in-deep-ice | Tabular | medium |
| spaceship-titanic | Tabular | medium |
| playground-series-s5e3 | Tabular | medium |
| playground-series-s4e1 | Tabular | medium |
| playground-series-s4e12 | Tabular | medium |
| march-machine-learning-mania-2025 | Tabular | medium |
| playground-series-s4e2 | Tabular | medium |
| playground-series-s4e7 | Tabular | medium |
| playground-series-s4e3 | Tabular | medium |
| playground-series-s3e24 | Tabular | medium |
| playground-series-s4e10 | Tabular | medium |
| open-problems-single-cell-perturbations | Tabular | medium |
| playground-series-s5e5 | Tabular | medium |
| champs-scalar-coupling | Tabular | medium |
| playground-series-s5e2 | Tabular | medium |
| Binary Prediction of Poisonous Mushrooms | Tabular | medium |
| hms-harmful-brain-activity-classification | Tabular | medium |
| ventilator-pressure-prediction | Tabular | medium |
| predict-volcanic-eruptions-ingv-oe | Tabular | medium |
| smartphone-decimeter-2022 | Tabular | hard |
| h-and-m-personalized-fashion-recommendations | Tabular | hard |
| child-mind-institute-problematic-internet-use | Tabular | hard |
| stanford-ribonanza-rna-folding | Tabular | hard |
| home-credit-credit-risk-model-stability | Tabular | hard |
| stanford-covid-vaccine | Tabular | hard |
| equity-post-HCT-survival-predictions | Tabular | hard |
| santa-2023 | Tabular | hard |
| ai-village-capture-the-flag-defcon31 | Tabular | hard |
| fide-google-efficiency-chess-ai-challenge | Tabular | hard |
| playground-series-s3e18 | Tabular | hard |
| spooky-author-identification | Text | easy |
| ai-mathematical-olympiad-prize | Text | easy |
| detecting-insults-in-social-commentary | Text | easy |
| billion-word-imputation | Text | easy |
| Multi-Lingual Abilities | Text | easy |
| text-normalization-challenge-english-language | Text | easy |
| Shopping Knowledge Reasoning | Text | easy |
| text-normalization-challenge-russian-language | Text | easy |
| Understanding Shopping Concepts | Text | easy |

| Competition | Data Modality | Task Difficulty |
|---|---|---|
| wsdm-cup-multilingual-chatbot-arena | Text | medium |
| uspto-explainable-ai | Text | medium |
| llm-detect-ai-generated-text | Text | medium |
| learning-agency-lab-automated-essay-scoring-2 | Text | medium |
| llm-prompt-recovery | Text | medium |
| chaii-hindi-and-tamil-question-answering | Text | medium |
| facebook-recruiting-iii-keyword-extraction | Text | medium |
| llm-20-questions | Text | medium |
| jigsaw-toxic-comment-classification-challenge | Text | medium |
| jigsaw-unintended-bias-in-toxicity-classification | Text | medium |
| us-patent-phrase-to-phrase-matching | Text | medium |
| pii-detection-removal-from-educational-data | Text | medium |
| santa-2024 | Text | medium |
| lmsys-chatbot-arena | Text | medium |
| movie-review-sentiment-analysis-kernels-only | Text | medium |
| ai-mathematical-olympiad-progress-prize-2 | Text | medium |
| commonlit-evaluate-student-summaries | Text | medium |
| AI4Code | Text | medium |
| feedback-prize-english-language-learning | Text | medium |
| intent-recognition-on-low-resource-language | Text | medium |
| eedi-mining-misconceptions-in-mathematics | Text | hard |
| Next Product Title Generation | Text | hard |
| tensorflow2-question-answering | Text | hard |
| tweet-sentiment-extraction | Text | hard |
| google-quest-challenge | Text | hard |
| llms-you-cant-please-them-all | Text | hard |
| leaf-classification | Image | easy |
| dogs-vs-cats-redux-kernels-edition | Image | easy |
| dog-breed-identification | Image | easy |
| rsna-2023-abdominal-trauma-detection | Image | easy |
| plant-seedlings-classification | Image | easy |
| aerial-cactus-identification | Image | easy |
| arc-prize-2024 | Image | easy |
| paddy-disease-classification | Image | easy |
| bms-molecular-translation | Image | easy |
| invasive-species-monitoring | Image | easy |
| denoising-dirty-documents | Image | easy |
| herbarium-2020-fgvc7 | Image | medium |
| rsna-2022-cervical-spine-fracture-detection | Image | medium |
| tgs-salt-identification-challenge | Image | medium |
| cassava-leaf-disease-classification | Image | medium |
| herbarium-2021-fgvc8 | Image | medium |
| aptos2019-blindness-detection | Image | medium |
| hotel-id-2021-fgvc8 | Image | medium |
| siim-isic-melanoma-classification | Image | medium |
| uw-madison-gi-tract-image-segmentation | Image | medium |
| plant-pathology-2021-fgvc8 | Image | medium |
| herbarium-2022-fgvc9 | Image | medium |
| inaturalist-2019-fgvc6 | Image | medium |
| petfinder-pawpularity-score | Image | medium |
| statoil-iceberg-classifier-challenge | Image | medium |
| kuzushiji-recognition | Image | medium |
| seti-breakthrough-listen | Image | medium |

| Competition | Data Modality | Task Difficulty |
|---|---|---|
| imet-2020-fgvc7 | Image | medium |
| alaska2-image-steganalysis | Image | medium |
| ranzcr-clip-catheter-line-classification | Image | medium |
| plant-pathology-2020-fgvc7 | Image | medium |
| iwildcam-2020-fgvc7 | Image | medium |
| histopathologic-cancer-detection | Image | medium |
| rsna-miccai-brain-tumor-radiogenomic-classification | Image | hard |
| vinbigdata-chest-xray-abnormalities-detection | Image | hard |
| rsna-breast-cancer-detection | Image | hard |
| iwildcam-2019-fgvc6 | Image | hard |
| image-matching-challenge-2024 | Image | hard |
| siim-covid19-detection | Image | hard |
| osic-pulmonary-fibrosis-progression | Image | hard |
| vesuvius-challenge-ink-detection | Image | hard |
| whale-categorization-playground | Image | hard |
| cdiscount-image-classification-challenge | Image | hard |
| google-research-identify-contrails-reduce-global-warming | Image | hard |
| hubmap-kidney-segmentation | Image | hard |
| 3d-object-detection-for-autonomous-vehicles | Image | hard |
| WhoIsWho-IND | Graph | easy |
| Graph based Recommendation | Graph | easy |
| PMLDL | Graph | easy |
| predict-ai-model-runtime | Graph | medium |
| PST | Graph | medium |
| OGB-LSC-MAG240M | Graph | medium |
| OGB-LSC-PCQM4M | Graph | hard |
| AQA | Graph | hard |
| OGB-LSC-WikiKG90M | Graph | hard |
| Marinexplore and Cornell Whale Detection Challenge | Audio | easy |
| the-icml-2013-whale-challenge-right-whale-redux | Audio | easy |
| mlsp-2013-birds | Audio | easy |
| ml2021spring-hw2 | Audio | medium |
| freesound-audio-tagging-2019 | Audio | medium |
| tensorflow-speech-recognition-challenge | Audio | medium |
| Rainforest Connection Species Audio Detection | Audio | medium |
| bengaliai-speech | Audio | medium |
| Cornell Birdcall Identification | Audio | hard |
| BirdCLEF 2021 - Birdcall Identification | Audio | hard |
| birdclef-2024 | Audio | hard |
| User Behavior Alignment | Multimodal | easy |
| All-Around | Multimodal | easy |
| multi-modal-gesture-recognition | Multimodal | easy |
| nomad2018-predict-transparent-conductors | Multimodal | medium |
| NextProductPrediction | Multimodal | medium |
| nfl-player-contact-detection | Multimodal | medium |
| random-acts-of-pizza | Multimodal | medium |
| lux-ai-season-3 | Multimodal | medium |
| ariel-data-challenge-2024 | Multimodal | hard |
| leash-BELKA | Multimodal | hard |
| planttraits2024 | Multimodal | hard |
| drawing-with-llms | Multimodal | hard |

