# OpenReview forum: "Towards Adaptive ML Benchmarks: Web-Agent-Driven Construction, Domain Expansion, and Metric Optimization"
_ICLR.cc/2026/Conference — Submitted to ICLR 2026_

### Official Review · Reviewer_ghri · 2025-10-21

**Soundness:** 2
**Presentation:** 3
**Contribution:** 3
**Rating:** 6
**Confidence:** 2

**Summary:**

This paper proposes TAM Bench, a diverse, realistic, and structured benchmark for evaluating LLM-based agents on end-to-end ML tasks. TAM Bench features three key innovations: (1) A browser automation and LLM-based task acquisition system that automatically collects and structures ML challenges; (2) A leaderboard-driven difficulty modeling mechanism that estimates task complexity using participant counts and score dispersion, enabling scalable and objective task calibration; (3) A multi-dimensional evaluation framework.

**Strengths:**

1. Automation and Scalability: The Web-Agent-driven task acquisition method improves task collection efficiency.
﻿
2. Objective Difficulty Modeling: The leaderboard-based difficulty assessment is more objective and scalable than previous manual time estimates.
﻿
3. Enhanced Benchmark Diversity: The Full version offers significantly broader coverage across data modalities and application domains.
﻿
4. Comprehensive Multi-Dimensional Evaluation: The inclusion of Constraint Adherence and Format Compliance metrics effectively addresses the limitations of single-metric evaluations in existing benchmarks.

**Weaknesses:**

The evaluation relies on an LLM (e.g., GPT-4) as the judge. The paper, however, does not discuss whether LLM-based evaluation can faithfully and objectively reflect the true capabilities of the models. It is suggested that necessary experiments be added to demonstrate (1) the gap between LLM evaluation and human evaluation, (2) the reliability of different LLM judges, and (3) whether GPT-4 can be replaced by an open-source model, especially given the relatively high cost of calling the GPT-4 API.

**Questions:**

Please see the weakness

---

### Official Review · Reviewer_zzJz · 2025-10-23

**Soundness:** 3
**Presentation:** 2
**Contribution:** 2
**Rating:** 2
**Confidence:** 4

**Summary:**

In this work, the authors aim to address several limitations of existing agent benchmarks—such as high manual annotation cost, imbalanced task distribution, and poorly calibrated task difficulty. To this end, they propose TAM-Bench, a diverse, realistic, and well-structured benchmark for evaluating LLM-based agents on end-to-end machine learning tasks. While the benchmark demonstrates clear advantages in terms of task diversity and scale, there remain notable shortcomings in the overall framework of its construction and evaluation methodology.

**Strengths:**

1. The task scale is 150, much larger than existing benchmarks such as MLEBench (75 tasks).
2. The proposed benchmark contains more task fields like commerce, which is important for real-scenarios.

**Weaknesses:**

1. In this work, the authors propose a difficulty modeling method via leaderboard structure, with many details unclear and questionable.
(1) Since they use the score from the participants to determine the task difficulty, is there any filter mechanism on the participants? If no, how to avoid the distribution shift led by the difference of participants?
(2) Current inclusion of number of participants seems not reasonable. Is there any scene that one task is too difficult / heavy to run such that its number of participants would be 1/100 or even 1/1000 of other simple-to-run tasks? In such case, will the difficulty be influenced in a wrong way?
(3) Given all factors except the "mean score" fixed in eq (3), we might conclude that the higher the mean score is, the more difficult the task is, which is not reasonable.

2. While the format validity metric is reasonable to evaluate the performance of agents, I think previous benchmarks might in-explicitly consider it, i.e., if it does not follow to the format, its answer might not even be parsed. Furthermore, I would appreciate it if the authors would provide more details of the generation of format requirements: test_labels.csv. If it is inherit from the construction of the task, I wonder its validness and diversity to evaluate agents' capability on this.

3. The evaluation of this benchmark is not sufficient. Only GPT-4.1 & Deepseek-V3 are tested, and their performance seems different from the common sense knowledge on these two models. Further analyses are expected.

4. Please adjust the usage of \cite, \citep, \citet in the latex.

**Questions:**

See Weaknesses,

---

### Official Review · Reviewer_HQoU · 2025-10-30

**Soundness:** 2
**Presentation:** 1
**Contribution:** 2
**Rating:** 2
**Confidence:** 4

**Summary:**

The paper introduces TAM-Bench, a new benchmark designed to test how well LLM-based agents can handle end-to-end machine learning tasks. Instead of relying on manual curation, it automatically gathers and standardizes real competition tasks from sites like Kaggle using a web-agent system. It also estimates task difficulty from leaderboard data and evaluates agents across several aspects, including performance, constraint following, and output format correctness. In experiments with AIDE and OpenHands using GPT-4.1 and DeepSeek-V3, GPT-4.1 was generally more stable and reliable, while DeepSeek-V3 showed strong results on certain tasks. Overall, TAM-Bench aims to provide a more practical and scalable way to evaluate AutoML agents in realistic settings.

**Strengths:**

1.	The paper presents an automated and scalable benchmark pipeline that reduces manual effort and ensures diverse task coverage.
2.	The leaderboard-based difficulty modeling offers a more objective and reproducible way to assess task complexity.
3.	The evaluation framework is comprehensive, considering both performance and practical constraints.

**Weaknesses:**

1.	The experimental design is shallow. TAM-Bench evaluates two open-source AutoML agent frameworks, but each framework’s base language model includes only one open-source model (DeepSeek-V3) and one closed-source model (GPT-4.1). Evaluating only two models is far from comprehensive and cannot reflect the capability boundaries of diverse AutoML agents, offering limited value to the community.
2.	The selection of base models is arbitrary. Excluding the Qwen series models simply because they “encountered JSON parsing errors during execution” is unreasonable, as this issue could be resolved through function calling or post-processing the responses. Furthermore, it is unclear why the authors chose DeepSeek-V3 instead of Llama-3 or other comparable language models.
3.	The authors propose an automatic pipeline for benchmark construction, but they do not systematically discuss the quality of the synthesized data, nor do they conduct any manual quality inspection of the benchmark samples. I am seriously concerned about the reliability of the automatically generated data.
4.	The writing is poor. For example, Figure 1 is never mentioned in the main text, and its caption fails to provide any meaningful information, which leaves readers confused.

**Questions:**

1.	TAM-Bench focuses on language model-based agents, so how does it handle inputs such as audio and images?
2.	The evaluation metrics in TAM-Bench are all based on final submissions, yet in long-sequence agent tasks, assessing the intermediate process is also meaningful. Why does TAM-Bench only consider result-based metrics?

---

### Meta-Review · Area_Chair_tsE8 · 2026-01-07

**Summary:**

The author did not submit rebuttal. After carefully checking the paper, the reviews, I think the weak points outweight the strong points. The author didn't address any concerns of reviewers. So I suggest reject this paper.

**Reviewer Concerns:**

No concerns have been addressed by the rebuttal.

**Reviewer Scores:**

The score remains unchanged.

---

### Decision · Program_Chairs · 2026-01-26

Reject